# AUTOTOOL: AUTOMATIC SCALING OF TOOL-USE CAPABILITIES IN RL VIA DECOUPLED ENTROPY CONSTRAINTS

**Yirong Zeng[1], Xiao Ding[1]\*, Yufei Liu[2], Yuxian Wang[3], Qunyao Du[1], Yutai Hou[3]\***
Wu Ning[3], Haonan Song[3], Duyu Tang[3], Dandan Tu[3], Bing Qin[1], Ting Liu[1]

**[1]Harbin Institute of Technology, SCIR Lib**
[2]Peking University
[3]Huawei Technologies Ltd.

**\*Corresponding authors**
`{yrzeng,xding}@ir.hit.edu.cn, houyutai@huawei.com`

## ABSTRACT

Tool use represents a critical capability for AI agents, with recent advances focusing on leveraging reinforcement learning (RL) to scale up the explicit reasoning process to achieve better performance. However, there are some key challenges for tool use in current RL-based scaling approaches: (a) direct RL training often struggles to scale up thinking length sufficiently to solve complex problems, and (b) scaled-up models tend to overthink simpler problems, resulting in substantial token inefficiency. To address these challenges, we propose a novel training paradigm that first employs warm-up supervised fine-tuning to help models distinguish between simple and complex problems, followed by RL that enable models to automatically determine appropriate reasoning trajectories. Furthermore, to tackle the issue of automatic thinking-length scaling, we discover that entropy-based optimization objectives effectively maintain model diversity while successfully unlocking the model's scaling capabilities. Based on this insight, we introduce an entropy-based long-short reasoning fusion RL strategy. Our experiments on three benchmarks demonstrate that model successfully achieves auto-scaling for efficient tool use, achieving significant 9.8% accuracy improvements while reducing computational overhead by ~81%.

## 1 INTRODUCTION

Integrating agentic large language models (LLMs) with external tools has emerged as a pivotal advancement, and has become a defining feature of advanced agentic models (OpenAI, 2025; K2, 2024). It significantly enhances a model's ability to address complex tasks (Qu et al., 2025; Wang et al., 2024), and opens up many practical uses across different fields. For example, it supports the automation of reasoning tasks (Jin et al., 2025; Li et al., 2025b), and enables agent applications (Zihan et al., 2025; Ouyang et al., 2025). Therefore, research on agentic tool use represents a critical pathway toward artificial general intelligence. In this task, models respond to queries by dynamically selecting and invoking relevant tools from an available pool.

Test-time scaling (TTS) is a approach to language modeling that uses extra test-time compute to improve performance (Muennighoff et al., 2025). Currently, scaling up a model's explicit reasoning length via Reinforcement Learning with Verifiable Rewards (RLVR) is a effective to achieve TTS. Compared to the prevalent Supervised Fine-Tuning (SFT) approach, which imitates reasoning patterns from labeled high-quality examples to teach models using tools (Liu et al., 2024; Zhang et al., 2024a), RLVR better fosters intrinsic reasoning, rather than making models memorize training trajectories (Chen et al., 2025). It has demonstrated robustness in mathematics (Shao et al., 2024) and

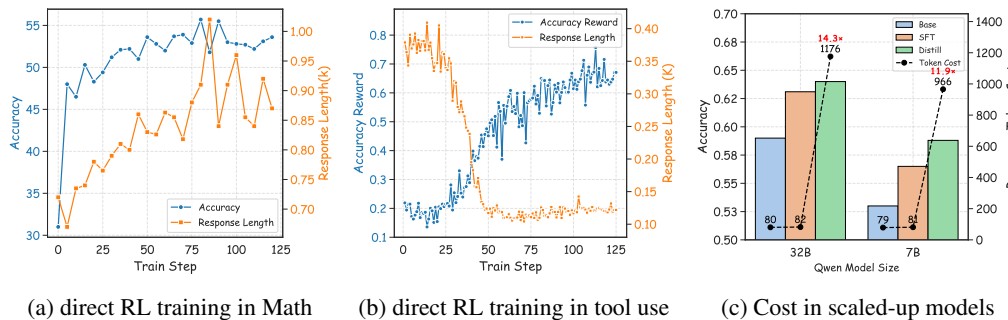

(a) direct RL training in Math    (b) direct RL training in tool use    (c) Cost in scaled-up models

Figure 1: The training paradigms for TTS in tool-use: (a) direct RL enables scaling up response length as accuracy improves in mathematical tasks; but (b) it fails to scale in tool-use tasks, where reasoning collapses into short trajectories; (c) scaled-up models (e.g., distillation models) incur significant token costs, as they require lengthy reasoning trajectories for all queries.

coding (Pan & Liu, 2025), while also driving a paradigm shift from SFT to RL in LLM training. Thus, exploring suitable scaling strategies is critical to advance effective agentic tool-use.

To this end, we pre-analyze the training paradigm for TTS in tool use, as shown in Figure 1. Under direct RL training, contrary to mathematical tasks where response length scales with improving accuracy, we observe that models suffer from *reasoning collapse* in tool use, a phenomenon where models fail to sufficiently extend thinking[1] length to solve complex problems. More importantly, we find many tool-use problems can be solved with short reasoning trajectories, yet scaled-up models generate excessively long trajectories that cause unnecessary resource consumption. Therefore, an adaptive model that dynamically integrates short and long reasoning is highly desirable.

More analysis in Section 2 reveal that low entropy, which quantifies a model's response certainty and exploration capability, leads to insufficient reasoning length, limiting the robustness of LLMs in tackling hard problems. These findings motivate our proposed method, which decouples long and short reasoning to prevent dominance interference while incorporating entropy constraints to enable long-trajectory reasoning. Therefore, we propose a decoupled adaptive entropy constraint strategy for RL. It first performs warm-up using a constructed mixed dataset of long and short reasoning trajectories to perceive data difficulty. The strategy decouples the policy loss between short and long reasoning, then applies varying entropy constraint strengths to regulate the thinking mode while maintaining concise responses for simple problems. This enables differentiated exploration control across reasoning modes, with adjusted entropy strength set above a target entropy in long reasoning to preserve exploration capacity.

Experiments on three benchmarks show: (1) Our ~7B model leads at comparable size models (e.g., +11.95% compare to SFT-model). (2) Beyond performance, our auto-scaling model boosts accuracy by 9.8% compared to the distilled model and cuts inference token cost by ~81%. Notably, our model's thinking rate reaches 45% in complex scenarios but 0% in simple ones. Moreover, visualizations of the training process confirm that our approach generates concise responses for simple cases while extending reasoning trajectories by $5\times$ for complex questions. This contrast demonstrates that the model has learned to automatically adjust test-time scales based on sample difficulty, ultimately supporting improved inference efficiency.

## 2 PRELIMINARY STUDY

In this section, we present extensive experiments to highlight the challenges of achieving test-time scaling for agentic tool use, and thereby motivate our method.

---

[1]In this paper, the terms *thinking* and *long reasoning* are used interchangeably, both referring to responses that contain an explicit reasoning process.

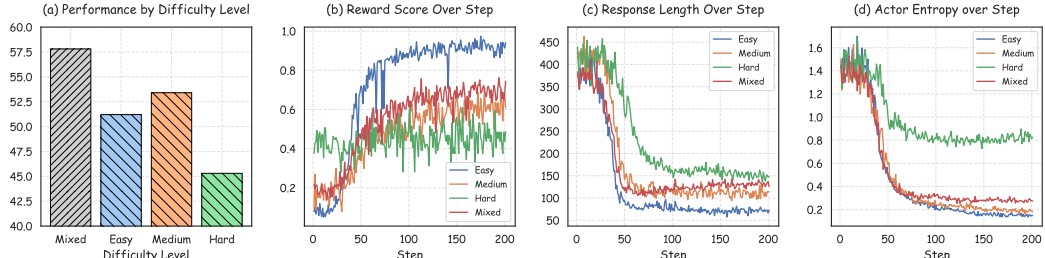

Figure 2: Impact of difficulty distributions. *Easy* and *Medium* converged successfully, while *Hard* failed (a, b). However, collapse occurred across all three subsets (c), with the same trend observed in entropy (d). This indicates that data distribution has no correlation with collapse, whereas low entropy exhibits a strong positive correlation.

## 2.1 TASK OVERVIEW

In agentic tool use, the LLM receives a user query $q$ along with candidate tools, represented as $\mathcal{T} = \{t_0, t_1, \ldots, t_{|\mathcal{T}|}\}$. The purpose of LLM is to fulfill the user's intent by executing a specific sequence of tools. We formalize this decision-making process as $y \sim \pi(y \mid q, s, \mathcal{T})$, where $\pi(\cdot)$ represents the policy model, $s$ denotes the task state (e.g., historical context ), and $y$ represents the actions taken by the model, such as selecting or executing tool calls. A review of related work is provided in Appendix A.

## 2.2 TRAINING PARADIGMS ANALYSIS

We analyze training paradigms for scaling reasoning process in tool use, including RL training and SFT with distillation, using Qwen2.5-series models to conduct training on the public *ToolACE* dataset (Liu et al., 2024) and evaluation via *BFCL* (Yan et al., 2024) (details in Section 4.1).

(1) For direct RL, we applied RL (specifically GRPO (Shao et al., 2024)). As shown in Figure 3b, we observe a divergence between the model's performance and response length: as training steps increased, performance improved while response length decreased sharply. This indicates the reasoning patterns collapsed into short reasoning trajectories, failing to scale-up in test-time. This result contradicts widely accepted findings from training on complex reasoning tasks (e.g., mathematics) (Hugging Face, 2025; Zeng et al., 2025b), as shown in Figure 3a, where we adopt experimental results from Zeng et al. (2025a). The evaluation results presented in Table 1 indicate that performance in complex tool-use scenarios (e.g., Multi-Turn) decreased noticeably compared to the distilled SFT model. This indicates that the reasoning collapse phenomenon limits the model's robust performance on complex problems.

(2) For SFT with distillation, we conducted base SFT and distillation from reasoning LLM (DeepSeek-AI, 2025b), respectively. As shown in Figure 1c, distilled models showed no noticeable accuracy gain over the base SFT, but increased output token costs by more than $10\times$. This suggests that many agentic tool-use problems can be solved with short reasoning trajectory while excessive long trajectory leads to unnecessary resource consumption.

## 2.3 PRE-STUDY ON REASONING PATTERN COLLAPSE.

To investigate the causes of reasoning pattern collapse, we conducted an in-depth analysis of data difficulty distribution and information entropy.

### 2.3.1 DATA DISTRIBUTION

Intuitively, we hypothesized that the sample difficulty distribution might exert a critical influence. Guided by this hypothesis, we used a base model to perform 8 rounds of reasoning on the training data and calculated pass@8. The resulting distribution (shown in Appendix B) reveals that **easy** samples (with $8/8$ correct inferences) and **hard** samples (with $0/8$ correct inferences) accounted for 47% and 31.8% of the dataset, respectively, while **medium** samples made up a smaller proportion.

Table 1: Evaluation results on the BFCL benchmark, which includes three sub-metrics: Non-live, Live, and Multi-Turn (including multi-turn and long-context tool-use scenarios). $l$ denotes target response length in response constraint, and $\beta$ denotes coefficients of entropy constraint.

| Model | Think? | Non-Live | Live | Multi-Turn | Overall Acc |
|---|---|---|---|---|---|
| Base LLM | ✗ | 86.46 | 67.44 | 7.62 | 53.69 |
| w/ SFT | ✗ | 86.65 | 75.11 | 6.75 | 56.90 |
| w/ distilled SFT | ✓ | 87.35 | 79.59 | 16.95 | 59.23 |
| w/ GRPO | ✓ | 87.06 | 78.22 | $8.38_{\downarrow 8.57}$ | 57.81 |
| w/ length constraint | | | | | |
| $+l = 100$ | ✓ | 87.30 | 71.23 | 7.92 | 55.37 |
| $+l = 50$ | ✓ | 87.76 | 78.43 | 8.78 | 58.12 |
| $+l = 10$ | ✓ | 89.76 | 77.33 | 8.89 | 58.27 |
| w/ entropy constraint | | | | | |
| $+\beta = 1e{-}2$ | ✓ | 87.47 | 79.13 | 9.48 | 59.33 |
| $+\beta = 5e{-}2$ | ✓ | 87.21 | 77.96 | 10.02 | 58.91 |
| $+\beta = 1e{-}1$ | ✓ | 88.32 | 80.42 | $15.86_{\uparrow 7.48}$ | 61.86 |

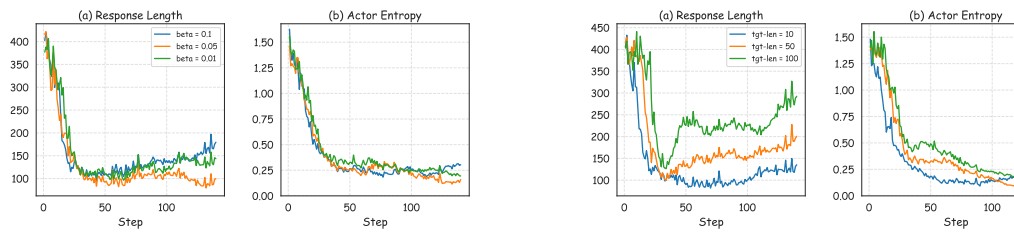

(a) RL with entropy constraint          (b) RL with length penalty

Figure 3: Training dynamics visualized for entropy constraint (a) and length penalty (b). The entropy constraint partially increases response length, yet the length penalty does not mitigate low entropy.

We then conducted separate RL training runs on these three subsets, evaluated the resulting models, and reported their training dynamics in Figure 2. Notably, reasoning pattern collapse persisted across all three subsets: after an initial exploration phase, easy samples led to rapid convergence, intermediate samples resulted in slower convergence, and hard samples showed no convergence. These findings indicate that the sample difficulty distribution can slightly reduce convergence speed, but no correlation with collapse has been observed.

The actor model's information entropy quantifies its exploration capability during training. As shown in Figure 2d, entropy decreases rapidly, with dynamics closely aligning with reasoning pattern collapse. Additionally, comparing three subsets, the final converged entropy increases sequentially from simple to complex samples. This reveals: simple problems elicit high certainty in short reasoning (perceiving extended exploration risks suboptimal solutions); complex problems face inherent challenges, with short reasoning advantage dominance further discouraging exploration and driving default to brief responses. This finding demonstrates a strong positive correlation between low entropy and collapse.

### 2.3.2 Information Entropy Constraints

To explore reasoning collapse-entropy connections, we incorporated entropy constraints into the policy loss function. Inspired by He et al. (2025), we designed a mechanism to maintain the entropy ($e$) at a reasonable level throughout training. The entropy loss is defined as:

$$loss_k^e = \beta \cdot \mathbb{I}\{e_k \leq \texttt{tgt-ent}\} \tag{1}$$

where $k$ is the training step, $\beta$ is the coefficient, and we set $\texttt{tgt-ent}=0.1$. Notably, the entropy loss is only activated when $e_k \leq \texttt{tgt-ent}$, ensuring the model's entropy remains lower-bounded by the target value. For comparative purposes, we also implemented a short-response penalty by configuring the reward function to penalize response below the target length $l$. The evaluation results

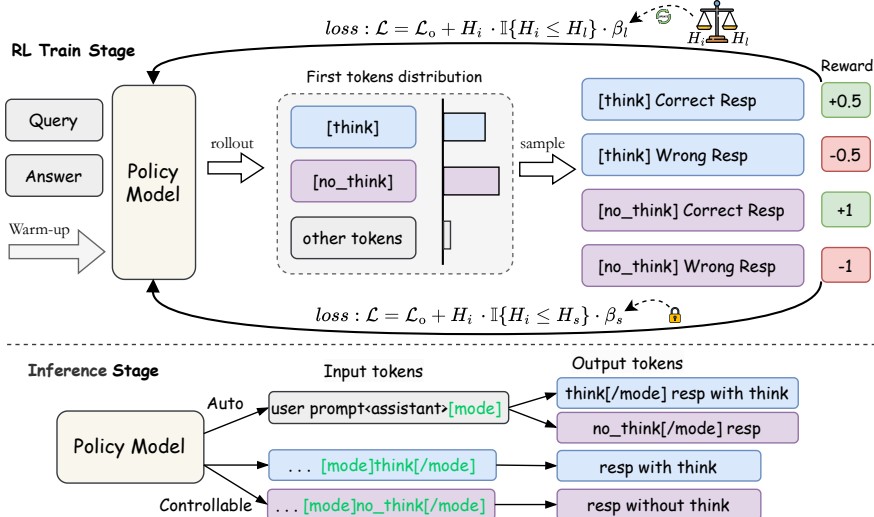

Figure 4: The overview of decoupled adaptive entropy constraint. It achieves automatic scaling by decoupling different reasoning modes through the application of differentiated entropy constraints. Adaptive entropy constraint strength for long reasoning. During the inference, the model can automatically or controllably switch inference modes by pre-pending a response prefix in Input tokens.

(presented in Table 1) show that the length constraint did not improve the model's test performance. In contrast, the entropy constraint yielded partial performance gains (visualizing the training process in Figure 3). However, the effectiveness is highly sensitive to $\beta$: in multi-turn, the maximum positive gain was achieved when $\beta = 1e-1$, whereas results for other $\beta$ are comparable to *w/ length constraints*. This sensitivity highlights the difficulty of pre-selecting an optimal entropy coefficient, indicating that dynamically adjusting $\beta$ during training is necessary.

Therefore, we propose a novel strategy: decoupled adaptive entropy constraints. It decouples entropy constraint for short and long reasoning and adaptively tunes the entropy coefficient, addressing both the collapse caused by low entropy and the sensitivity of static coefficients.

## 3 METHODOLOGY

Our method first performs warm-up SFT to perceive sample difficulty (details in 3.1), followed by RLVR training with decoupled adaptive entropy constraints, as shown in Figure 4.

### 3.1 DATA PREPARATION AND WARM-UP

To support robust general-purpose tool use via RL, we constructed a mixed dataset covering diverse tool-use scenarios from public sources: ToolACE (Liu et al., 2024), xLAM (Zhang et al., 2024a; Prabhakar et al., 2025), Hermes Function-Calling (interstellarninja, 2024). More details are provided in Appendix B. To create a balanced dataset encompassing diverse complexity levels and tool usage scenarios, we randomly downsampled the raw data. Moreover, we adopted the following strategies to develop a Public agentic Tool-use dataset (**PubTool**), as presented in Table 2.

**Warm-up Training.** To help the model initially perceive data difficulty, we propose SFT for warm-up training by mixing long and short reasoning data. To construct such mixed thinking data, we performed multiple inferences (calculating pass@8) on the training data using Qwen2.5-7B-Instruct (no-thinking model) and Qwen3-32B (thinking model), respectively. For each response turn, we adopted the ground truth as the label if the no-thinking model's output was correct; otherwise, we adopted the thinking model's answer with explicit long reasoning if it was correct. More Details of data preparation are shown in Appendix §B. We design an auto-thinking template (details in Appendix §E) to enable the model to select reasoning modes based on data difficulty. Finally, we conducted SFT on the base model for warm-up, preparing for subsequent RL scaling.

Table 2: Data statistics of **PubTool** in data collection and construction. Subscript text in the SFT data table indicates the thinking rate in all turns.

| | ToolACE | xLAM | Hermes Function-Calling |
|---|---|---|---|
| Raw Data | 11.3k | 65k | 7.1k |
| Downsampled | 11.3k | 15k | 7.1k |

| | **PubTool** | |
|---|---|---|
| Processed | SFT data | RL data |
| | 8.2k$_{(9.2\%)}$ | 7k |

**Quality Refinement for RL data**. To efficiently support auto-scaling RL training, we employed the following data enhancement strategies: First, from data distribution analysis in Section §2.3.1, we observed that the original dataset was dominated by overly simple and excessively difficult samples. Overly simple samples offer limited value for RL training and lack generalization, while overly difficult samples either exceed model capabilities or contain noise. We therefore randomly removed half of both simple and difficult samples to balance the dataset distribution. Additionally, inspired by Li et al. (2025a), we prioritized training samples based on their alignment with model learning trajectories. Specifically, we performed multi-epoch GRPO training on all training data, computed changes in their reward scores, and calculated each sample's variance relative to the average reward. Lower variance indicated higher alignment. Through these processes, we downsampled the RL dataset from 21k to 7k samples. For a detailed analysis of its effects, please refer to Appendix B.

## 3.2 DECOUPLED ADAPTIVE ENTROPY CONSTRAINTS

To enable automatic scaling in agentic tool use, we propose a *decoupled adaptive entropy constraints* strategy for RLVR. The objective policy loss integrates the surrogate objective from native RLVR (e.g., GRPO) with a mechanism that: (1) decouples entropy regulation between short and long trajectories; (2) adaptively adjusts the entropy strength in long reasoning trajectories to preserve exploration capacity.

Specifically, let $\pi_\theta$ be the policy, $H_i = -\mathbb{E}_{a\sim\pi_\theta(\cdot|s_i)}[\log \pi_\theta(a|s_i)]$ is the entropy at step $i$, and $m_i \in \{0, 1\}$ an indicator variable: equals 1 if the action step is a short trajectory and 0 if it is a long trajectory. We apply decoupled entropy constraints based on policy model's response trajectory type: (1) $\beta_s$: fixed coefficient for short paths (to prevent excessive exploration), (2) $\beta_l$: adaptive coefficient for long paths (learned dynamically).

The sample-level policy loss is defined as:

$$\beta_i = \beta_s \cdot m_i \cdot \mathbb{I}\{H_i \leq H_s\} + \beta_l \cdot (1 - m_i) \cdot \mathbb{I}\{H_i \leq H_l\}, \tag{2}$$

$$\mathcal{L}_\text{p} = \frac{1}{N} \sum_{i=1}^{N} \left[ -\min\left(\rho_i \hat{A}_i, \text{clip}(\rho_i, 1 - \epsilon, 1 + \epsilon) \cdot \hat{A}_i\right) - \beta_i H_i \right], \tag{3}$$

where $\beta_i$ adapts the entropy penalty per sample, $\rho_i = \pi_\theta(a_i|s_i)/\pi_{\theta_\text{old}}(a_i|s_i)$, $H_l$ and $H_s$ denote target entropy of long reasoning and short reasoning, and $\hat{A}_i$ is the estimated advantage based on reward scores in Section 3.3. The key design is the *decoupling* of entropy weights via $m_i$, enabling distinct regularization strategies.

**Adaptive Entropy Coefficient Loss.** Entropy regularization is highly sensitive to the choice of coefficient, making it difficult to select an optimal coefficient in advance. This motivates a dynamic adjustment of the entropy loss coefficient. To automatically adjust the entropy strength for long trajectories, we introduce an adaptive loss that updates $\beta_l$ based on the deviation of actual entropy from a target level. The loss is computed only on steps belonging to long trajectories ($m_i = 0$) and is defined as:

$$\mathcal{L}_\beta^l = \frac{1}{\sum_j (1 - m_j)} \sum_{i=1}^{N} (1 - m_i) \cdot \beta_l \cdot (H_i - H_l), \tag{4}$$

where $H_l$ is a predefined target entropy. The coefficient $\beta_l$ is updated by minimizing $\mathcal{L}_\beta^L$: if $H_i < H_l$, $\beta_l$ increases to encourage exploration; if $H_i > H_l$, it decreases to suppress excessive randomness. In contrast, $\beta_s$ remains fixed during training.

### 3.3 AUTO THINKING REWARD MODULE

In this module, the model's output is evaluated using a rule-based reward (DeepSeek-AI, 2025a; Meng et al., 2025) to compute the estimated advantage for the objective loss $\mathcal{L}_p$. Specifically, for each question $q$, model generates $G$ completions $\{o_1, o_2, \ldots, o_G\}$ using $\pi_{\theta_{\text{old}}}$. This reward module combines format and answer rewards to score each completion.

**Format Reward.** The format reward $\mathcal{R}_{\text{format}}(o_i) \in \{0, 1\}$ evaluates whether the output adheres to the required structural template. We define two valid reasoning modes: *think* and *no-think*, each with strict syntactic constraints:

```
[mode]think[/mode][think]reasoning process here[/think]answer
[mode]no_think[/mode][no_think]\n[/no_think]answer
```

This design encourages explicit reasoning for complex problems via the *think* mode, while allowing direct generation for simple queries via *no-think*, reducing computational overhead. During the inference stage, controllable reasoning modes are achieved by prepending special tokens to the input, as depicted at the bottom of Figure 4.

**Answer Reward.** We check the correctness of the tool call by comparing it against the ground-truth annotation $y^*$. Tool-call outputs are parsed into structured dictionaries, enabling exact matching of both the function name and all required arguments. To encourage a balance between reasoning efficiency and accuracy, we design an asymmetric reward based on the mode (*think* or *no-think*):

$$\mathcal{R}_{\text{answer}}(o_i) = \begin{cases} +1.0, & \text{if } o_i = y^*, \textit{no-think}, \\ +0.5, & \text{if } o_i = y^*, \textit{think}, \\ -0.5, & \text{if } o_i \neq y^*, \textit{think}, \\ -1.0, & \text{if } o_i \neq y^*, \textit{no-think}, \end{cases} \tag{5}$$

It incentivizes short responses when they are correct, while encouraging long reasoning when mistakes occur, prompting more careful processing in uncertain scenarios.

## 4 EXPERIMENTS

### 4.1 EXPERIMENTAL SETUP

We use the open-source Qwen2.5-7B-Instruct as our base model. We compared four baseline types: **Base**, **SFT**-trained , API-based **Frontier**, and **RLVR**-trained models. Additionally, we compare the series of models trained on the base model using *PubTool* data. See Appendix D for more details.

**Evaluation Dataset**. The following benchmarks are used for evaluation: (1) **BFCL** (Yan et al., 2024) provides a comprehensive dataset comprising 4k+ instances (updating), consisting of *Non-live* (with expert-curated simple tools), *Live* (with user-contributed complex tools), *Multi-turn* (with multi-turn & multi-step tool use) samples. (2) **API-Bank** (Li et al., 2023), which consists of 314 tool-use dialogues and 753 API calls. This dataset evaluates a models' abilities to correctly invoke a known API (L-1) based on a query and to retrieve and call APIs from a tool list (L-2). (3) **ACEBench** (Chen et al., 2025) is a 2k-entry benchmark for assessing agentic tool use, using its summary score in "normal" evaluation type (covering single-turn and multi-turn scenarios).

### 4.2 OVERALL PERFORMANCE

The overall performance of models are shown in Table 3 and Figure 5. Firstly, the results indicate that our model consistently achieves corresponding best performance at comparable scales (~7B). For instance, compared to PubTool-SFT, AutoTool-7B with automatic think achieving +11.95 point improvement. And relative to Base model, it also has a remarkable boost with +16.43%. Secondly,

our model demonstrated its more superiority in challenging scenarios (e.g., achieves +28.5% improvement compare to PubTool-SFT in *Multi-turn*). This demonstrates that our method realizes a strong robustness enhancement in complex scenarios.

Moreover, our model outperforms most SFT-trained and RLVR-trained models in BFCL, and demonstrates comparable performance with the frontier models. It also shows consistent advantageous performance on API-Bank and ACEBench compared with baselines in Figure 5. For example, on ACEBench, our model achieves a 6.5 improvement compared to GRPO and a 5.9 improvement compared to Distilled. Finally, in the inference controllable mode, when forced to think, the overall performance is on par with auto think; when forced not to think, the effect on *Multi-turn* is significantly improved compared to no-think models (e.g., PubTool-SFT).

Table 3: Comparison on the BFCL benchmark. *Overall Acc* denotes the average performance on three subsets. [*] indicates a single-turn tool use model; [†] denotes models trained on *PubTool* data with a specific method. The subscript denotes the thinking rate.

| Type | Model | Non-Live | Live | Multi-Turn | **Overall Acc** |
|---|---|---|---|---|---|
| ♣Base | LLaMA-3.1-8B-Instruct | 84.21 | 61.08 | 9.62 | 50.87 |
| | Qwen2.5-7B-Instruct | 86.46 | 67.44 | 7.62 | 53.69 |
| | Qwen2.5-32B-Instruct | 85.81 | 74.23 | 17.75 | 59.67 |
| ♥Frontier | GPT-4o-2024-11-20 | 87.67 | 79.88 | 43.00 | 70.42 |
| | o3-2025-04-16 | 81.42 | 73.43 | 56.12 | 70.32 |
| | Gemini-2.5-Pro | 89.54 | 76.83 | 30.62 | 65.48 |
| ♦SFT | Hammer2.1-7b(Lin et al., 2024) | 88.65 | 75.11 | 23.50 | 61.83 |
| | ToolACE-8B (Liu et al., 2024) | 87.54 | 78.59 | 7.75 | 58.42 |
| | xLAM-7b-r(Zhang et al., 2024a) | 81.06 | 75.22 | 10.00 | 54.75 |
| | PubTool-SFT[†] | 88.98 | 77.28 | 9.68 | 58.17 |
| | PubTool-Distilled[†] | 87.73 | 78.64 | 15.65 | 60.30 |
| ♠RLVR | DeepSeek-R1-0528 | 75.20 | 77.30 | 38.88 | 63.79 |
| | Qwen3-8B(Team, 2025a) | 88.81 | 78.54 | 33 | 66.34 |
| | QwQ-32B(Team, 2025b) | 87.33 | 75.61 | 14.50 | 58.30 |
| | Tool-N1-7B[*](Zhang et al., 2025b) | 89.25 | 80.38 | - | - |
| | ToolRL-7B(Qian et al., 2025) | 82.21 | 74.90 | 18.12 | 58.38 |
| | Thinkless(Fang et al., 2025) | 86.92 | 77.62 | 24.64 | 63.06 |
| | Adactrl(Huang et al., 2025) | 86.36 | 73.12 | 15.63 | 58.37 |
| | PubTool-GRPO[†] | 88.87 | 78.93 | 10.77 | 60.13 |
| ♠Ours | AutoTool-7B[†] | $89.76_{0\%}$ | $80.22_{4.8\%}$ | $38.18_{45\%}$ | $70.12_{9.7\%}$ |
| | + *think* | 89.86 | 80.43 | 39.28 | 70.71 |
| | + *no-think* | 87.36 | 78.60 | 27.63 | 63.34 |

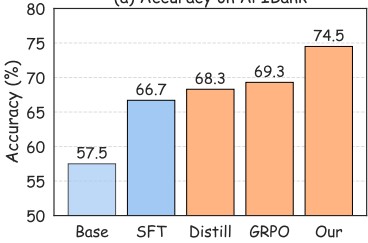
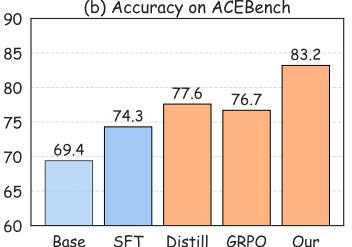

Figure 5: Performance of methods using training data *PubTool* on APIBank and ACEBench.

### 4.3 DEEP ANALYSIS STUDY

#### 4.3.1 ABLATION STUDY

To evaluate the effectiveness of key components in our method, we conducted an ablation study with the following variations: (1) Replaced the adaptive entropy coefficient with a fixed one (*w/o*

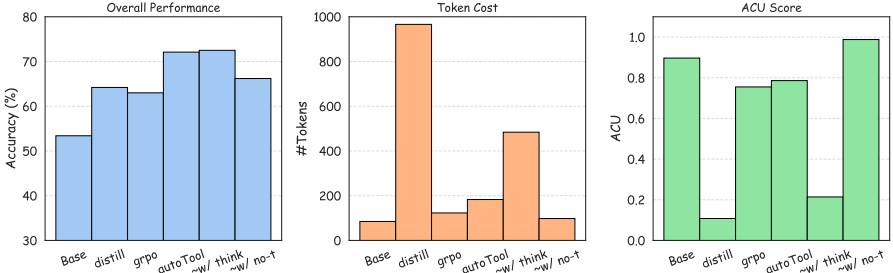

Figure 6: Inference efficiency analysis results, including performance, token cost, ACU.

*adapt coeff* ); (2) Replaced the decoupling loss with a unified loss with fixed entropy constraint (*w/o decouple*); (3) Removed data quality refinement (*w/o data refine*). We also included Qwen2.5-7B-Instruct as a Base Model for comparison. As shown in Table 4, compared with the baseline, our full model delivers a significant improvement of 16.43 points in Overall performance. All components are essential to our method, and removing any component causes clear performance drops: (1) *w/o data refine* brings the largest 6.43% Overall reduction, highlighting high-quality data as a core foundation. (2) *w/o adapt coeff* leads to a 10.53% Multi-turn decline, proving its value in stabilizing multi-round interactions; (3) *w/o decouple* results in a 2.34% Overall drop, showing decoupling avoids objective interference.

Table 4: The strategy ablation performance ($\uparrow$ = increase, $\downarrow$ = decrease, values are relative percentage changes from the *Our (w/. all)* model).

| Models | Non-live | Live | Multi-turn | Overall |
|---|---|---|---|---|
| Base Model | 86.46 | 67.44 | 7.62 | 53.69 |
| Our (w/. all) | 89.76 | 80.22 | 38.18 | 70.12 |
| w/o. *data refine* | 88.22 $_{\downarrow 1.54}$ | 73.29 $_{\downarrow 6.93}$ | 26.84 $_{\downarrow 11.34}$ | 63.69 $_{\downarrow 6.43}$ |
| w/o. *decouple* | 87.35 $_{\downarrow 2.41}$ | 75.98 $_{\downarrow 4.24}$ | 27.65 $_{\downarrow 10.53}$ | 64.23 $_{\downarrow 5.89}$ |
| w/o. *adapt coeff* | 88.73 $_{\downarrow 1.03}$ | 78.73 $_{\downarrow 1.49}$ | 32.14 $_{\downarrow 6.04}$ | 67.78 $_{\downarrow 2.34}$ |

### 4.3.2 INFERENCE EFFICIENCY ANALYSIS

Given the trade-off between reasoning path length, model size (~B), and performance, we introduce a new metric, Accuracy per Computation Unit (ACU), to better capture this balance and assess model inference efficiency (Ma et al., 2025). It is defined as:

$$\text{ACU} = \frac{\text{Accuracy}}{\#\text{Params} \times \#\text{Tokens}} \quad (6)$$

Since the ACU value typically falls within the range of $10^{-5}$ to $10^{-3}$, we report it in units of $10^3$ for improved readability. In addition, we report the thinking rates of our model across all submetrics.

The experimental results are summarized in Figure 6. From the results, we observe that AutoTool achieves the second-best overall performance: it reduces token cost significantly by 81%, requiring only about ~183 tokens compared to the distilled model (~966 tokens). Notably, with the forced no-think inference mode, AutoTool attains the optimal ACU score (0.97), demonstrating excellent inference efficiency. Even with the think inference mode, it still delivers the highest accuracy while cutting the token cost by half relative to the distilled model. Additionally, Table 3 shows that the our model's thinking rate reaches 45% in the Multi-Turn scenario but 0% in the No-Live scenario. The training process visualized in Appendix Section C shows our model extends reasoning trajectories for complex questions by $\times 5$, while enabling concise responses for simple ones. This suggests the model has learned to automatically adjust the test-time scale based on sample difficulty, which effectively supports the improvement of inference efficiency.

## 5 CONCLUSION

This study focused on addressing challenges in integrating agentic LLMs with tools by optimizing the RLVR paradigm. Our research first identified two critical issues: excessive resource consumption caused by unnecessary long-trajectory reasoning, and the *reasoning collapse* phenomenon under the direct RL training, hindering effective scaling. To solve these, we proposed a decoupled adaptive entropy constraint strategy, which enables the model to automatically adjust reasoning scales based on problem difficulty, thereby balancing performance and inference efficiency. Experiments on three benchmarks confirmed the strategy's effectiveness, boosting accuracy while cutting inference token cost significantly. This work advances RL-based agentic tool-use training and provides a practical auto-scaling solution for efficiently handling tasks.

## ACKNOWLEDGEMENTS

The research in this article is supported by the New Generation Artificial Intelligence of China (2024YFE0203700), National Natural Science Foundation of China under Grants U22B2059 and 62576124.

## ETHICS STATEMENT

This work strictly adheres to the ICLR Code of Ethics: it involves no human subjects, uses datasets compliant with original licensing agreements (ensuring privacy and legal compliance), and avoids discriminatory biases in experimental design/results; all authors confirm adherence, with no conflicting sponsorships.

## REPRODUCIBILITY STATEMENT

For reproducibility, key details are referenced across the main text (methodology Section 3.2, experimental setup Section 4.1), appendix (hyperparameters Section D, data processing details Section B, full prompt Section E), and supplementary materials (anonymous source code). We ensure data splits, random seeds, and environment configurations are explicitly stated, allowing researchers to independently verify our findings under identical conditions.

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

## USE OF LLM

LLMs (GPT-4o) were only used as general-purpose tools to draft baseline literature summaries and proofread minor grammar, no LLM contributed to core ideation, algorithm development, analysis, or writing, and all LLM-assisted content was verified for accuracy/integrity. No LLM is eligible for authorship.

## A RELATED WORK

### A.1 AGENTIC TOOL-USE

Enhancing LLMs with external tools has emerged as a pivotal direction for addressing complex tasks in open domains (Qu et al., 2025; Wang et al., 2024). Typical applications include integrating LLMs with search engines (Zhang et al., 2024b; Lazaridou et al., 2022; Shuster et al., 2022), calculators (Nakano et al., 2021), and Python interpreters (Wang et al., 2024; Song et al., 2024; Chen et al., 2022). Three common paradigms are widely adopted for training tool-use LLMs: (1) SFT: imitates the reasoning patterns from labeled high-quality examples, enabling models to learn standard tool-use workflows (Liu et al., 2024; Zhang et al., 2024a; Qin et al., 2023; Prabhakar et al., 2025). (2) RL with direct preference optimization: aligns model tool-use behavior with human intentions by optimizing against human preference signals (Zeng et al., 2025c; Yu et al., 2024). (3) RL with Verifiable Rewards (RLVR): as a novel approach, leverages scalable test-time inference and utilizes verifiable signals as rewards to refine the model's tool-use decisions (Li et al., 2025b).

## A.2 RL Scale-up

Reinforcement learning (RL) has gained traction as a more scalable and generalizable training paradigm. Models like R1-Zero leverage group relative policy optimization (GRPO) (Shao et al., 2024) to unlock the model's reasoning capabilities at test time (DeepSeek-AI, 2025a; Yu et al., 2025). This R1-style reasoning paradigm, marking a shift from train-time scaling to test-time scaling (Muennighoff et al., 2025; Xia et al., 2025), has demonstrated success in mathematics (Shao et al., 2024), coding (Pan & Liu, 2025), and agentic tool use (Feng et al., 2025; Jin et al., 2025).

Recently, several works have explored automatic scaling , i.e., enabling models to adaptively select the optimal reasoning mode based on problem difficulty (Fang et al., 2025; Zhang et al., 2025a; Huang et al., 2025; Wang et al., 2025). In agentic tool-use tasks, auto-scaling is particularly critical: many such problems can be solved with short reasoning, whereas excessively long reasoning leads to unnecessary resource consumption. While RL-based scaling for tool use in open-domain reasoning has been investigated (Zhang et al., 2025b; Qian et al., 2025), RL with auto-scaling remains unexplored in agentic tool use.

## B Details in Data Preparation

**Source of Training Data Details**. The raw data was sourced as follows:

- ToolACE (Liu et al., 2024): A general tool-use dataset teaching models when to invoke tools vs. respond directly, enhancing multi-step decision-making.
- xLAM (Zhang et al., 2024a; Prabhakar et al., 2025): A compositional dataset requiring one or more tool calls per turn. We mixed the original 60k xLAM with its multi-turn variant APIGen-MT-5k (Prabhakar et al., 2025).
- Hermes Function-Calling (interstellarninja, 2024): Designed to train LLMs in function calls and structured output from natural language. We extracted function call-related samples.

The dataset features various conversational scenarios where AI agents are required to interpret queries and execute appropriate single or multiple function calls. In Section 2, data distillation employs Deepseek-R1-0528 (DeepSeek-AI, 2025a). Subsequently, in Section §3.1, to mitigate model bias by aligning with a no-think model, data distillation is carried out using Qwen3-32B (Team, 2025a).

**Data Processing Pipeline & Distribution Details.** We obtained PubTool from raw data through following data processing workflow: (1) We randomly downsampled xLAM to balance the sample sizes across the three datasets. (2) We removed overly simple and excessively difficult samples; Figure 7 shows the raw-data distribution of successful reasoning counts (pass@8). Guided by this distribution, we partitioned the data into hard (31.8%), medium (21.2%), and easy (47%) subsets (Figure 7a) and re-balanced the difficulty distribution accordingly. (3) For RL data, we further refined the set by prioritizing samples that align closely with the model's current learning trajectory (details in the next paragraph). For comparison, we visualized the PubTool distribution in the same way (Figure 7b). We observed that the original corpus is concentrated in the easy and hard extremes, whereas PubTool peaks in the hard subset and is sparse in the easy subset. We argue that training on data of moderately high difficulty better elicits the model's test-time scaling capability (He et al., 2025).

**RL Data Refine Details.** In the second phase of data processing, we prioritize training samples by their alignment with model learning trajectories, measured through the variance of reward scores from the mean, lower variance indicates better alignment. Better alignment corresponds to lower variance of reward scores, defined as:

$$\text{Var}(r) = \frac{1}{n-1} \sum_{i=1}^{n} (r_i - \mu_r)^2, \quad \mu_r = \frac{1}{n} \sum_{i=1}^{n} r_i$$

where lower Var(r) indicates better alignment. This sampling result is illustrated in Figure 8. From the figure, we observe that the average reward ranges between 0.7 and 0.9, with aligned samples showing higher scores in the upper-left region and misaligned samples displaying lower scores. To

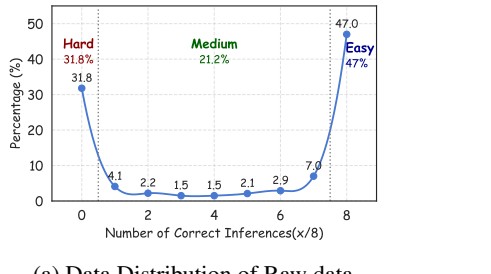 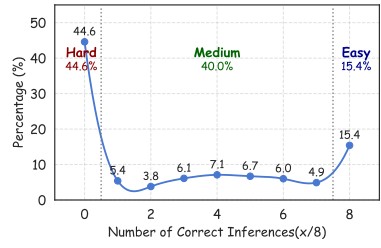

    (a) Data Distribution of Raw data              (b) Data Distribution of PubTool

Figure 7: The number of correct inferences distribution with performing 8 rounds of reasoning on the raw training data (a). The distribution of PubTool after data processing (b).

align the dataset size with that of SFT (≈8K samples), we rank examples by reward variance and filter out the bottom 55%, a simple yet effective heuristic to retain higher-signal instances while controlling scale.

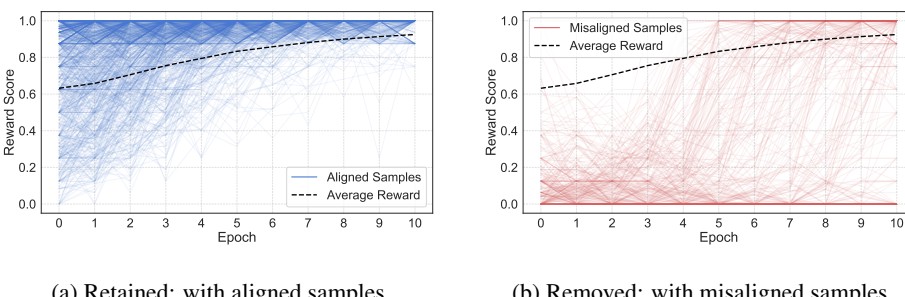

    (a) Retained: with aligned samples          (b) Removed: with misaligned samples

Figure 8: We retain aligned samples (i.e., those with low variance (a) ) and remove misaligned samples (i.e., those with high variance (b)).

**Effectiveness of Data Refinement**. We experimentally verified its effectiveness. After warm-up SFT, Figure 9 shows GRPO training processes with and without data refinement. Results indicate data refinement increases accuracy reward score by +15%, reduces training fluctuation variance, and enhances stability. Additionally, the model's thinking rate converged to a lower level, indicating improved memory capacity. BFCL evaluation results show GRPO with data refinement reached 66.82%, versus 60.78% without, an improvement of +6.04%. These enhancements are attributed to data refinement filtering substantial noise while retaining high-contribution samples.

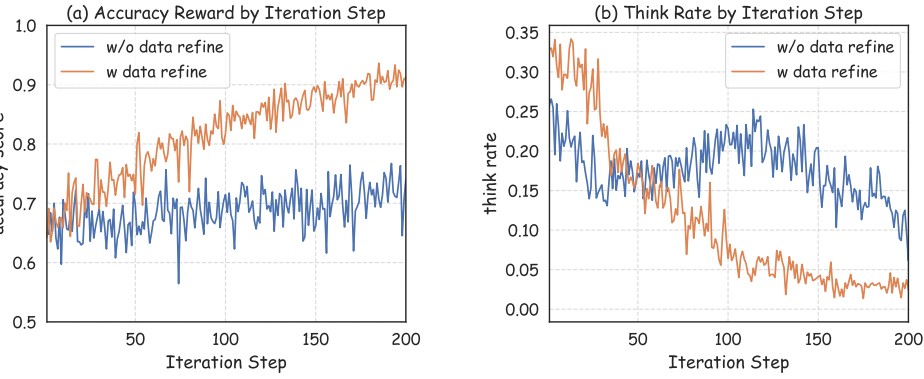

Figure 9: RL training processes (with and without data refinement) are shown, along with accuracy scores and thinking rates.

## C  VISUALIZATION OF TRAINING DYNAMICS

To demonstrate auto-scaling effects, we visualized the training process (Figure 10). As training progressed, accuracy improved while the thinking rate gradually decreased to 5%, indicating fewer problems required long reasoning, suggesting enhanced intrinsic tool-using capabilities. Additionally, response length and entropy achieved decoupled control: think mode enabled 500% longer reasoning trajectories than no-think mode, with corresponding higher actor entropy reflecting greater exploration tendency. These visualizations confirm that training enhanced tool-using abilities and successfully enabled auto test-time scaling based on problem difficulty and model proficiency.

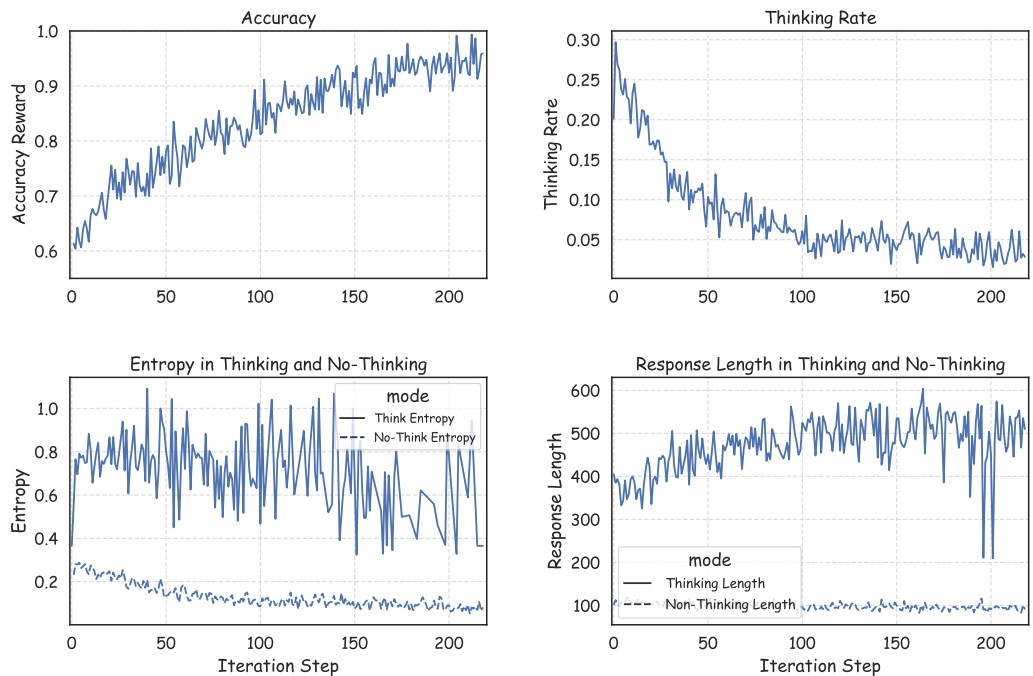

Figure 10: The visualization of training dynamics.

## D  COMPLEMENTARY EXPERIMENTS

### D.1  MORE IMPLEMENTATION DETAILS

The experiments were executed using the publicly accessible training framework MindSpeed-RL[2], an end-to-end reinforcement learning acceleration framework based on the Ascend ecosystem. The BFCL is an evolving benchmark and we utilized the version checked out on June 14, 2025. For the training model, we selected the best performance checkpoint on the valid dataset. In the Test-time Scale paradigms analysis (§2), we used an instruct model for SFT and a base model for GRPO. We employ a full-parameter training strategy for all SFT. In baseline trained on PubTool data, we trained on the complete dataset using specific methods (e.g., SFT, Distilled SFT, and GRPO). In the PubTool-GRPO training, we adopted the widely used think prompt pattern, which follows the format: `<think> reasoning process here </think><answer> answer here </answer>`. Each RL training run for the 7B model completed within 4 hours on a cluster of 32 Ascend 910b NPUs (configured as 4 nodes × 8 NPUs). The hyperparameters used are detailed in Table 5.

**Baselines** (1) *Base Model*: the original model without additional training (e.g., Qwen2.5-series, LLaMA3.1-series). (2) *SFT-trained Model*: ToolACE-8B (trained on the full ToolACE dataset (Liu

---
[2]https://gitee.com/ascend/MindSpeed-RL

et al., 2024)), xLAM-series (trained on the full xLAM dataset (Zhang et al., 2024a)), and Hammer-series (trained on xLAM with function masking (Lin et al., 2024)). (3) *API-based* closed-source frontier models (e.g., GPT-series, Gemini-series). (4) *RLVR-trained Model*: models trained using GRPO as the RL paradigm, such as QwQ-32B (Team, 2025b), Qwen3-series (Team, 2025a), Tool-N1 series (single-turn tool-use models trained on mixed ToolACE and xLAM data (Zhang et al., 2025b)), and ToolRL(trained in subset of mixed ToolACE and xLAM data (Qian et al., 2025)). Given that we are the first to investigate adaptive reasoning in agentic tool use, there is a significant lack of prior works on this domain, we compare our method with recent adaptive reasoning strategies adapted from other domains (e.g., math). We select two representative methods for comparison: (1) Thinkless (Fang et al., 2025): Achieves adaptive reasoning via a control token loss. (2) Adactrl (Huang et al., 2025): The model self-assesses problem difficulty during the RLVR rollout to facilitate adaptive reasoning. We replicated both methods using the same dataset (*PubTool*) and the same base model (Qwen2.5-7B-Instruct).

| Hyperparameter | Value | Hyperparameter | Value |
|---|---|---|---|
| **Data Configuration** | | **RL Optimization** | |
| Global Batch Size | 128 | Learning Rate | 1e-6 |
| Max Prompt Length | 12000 | LR Decay Style | constant |
| Max Response Length | 2048 | Mini Batch Size | 128 |
| Micro Batch Size | 4 | KL Loss Used | False |
| Train Steps | 200 | | |
| **Rollout Configuration** | | **Entropy Constraints** | |
| Rollout Name | vllm | Clip Higher $\epsilon$ | 0.28 |
| GPU Memory Utilization | 0.5 | Think Target Entropy $H_l$ | 0.2 |
| Number of Rollouts | 8 | No-Think Target Entropy $H_s$ | 0.1 |
| Temperature | 1.0 | Init Adaptive Coefficient $\beta_l$ | 0.1 |
| Tensor Model Parallel Size | 1 | Fixed Coefficient $\beta_s$ | 0.1 |
| Top_P | 1.0 | | |

Table 5: The configurations for RL training with GRPO.

## D.2 HYPERPARAMETER SENSITIVITY ANALYSIS

In this section, we conduct a hyperparameter sensitivity analysis to explore the effects of key parameters, including the target entropies ($H_s$ and $H_l$) and the entropy penalties ($\beta_s$ and initial $\beta_l$). We evaluated the performance on the BFCL benchmark, reporting the Overall Accuracy.

**Sensitivity to Target Entropy** Table 6 presents the sensitivity analysis with respect to the target entropy values. The results indicate that model performance is sensitive to the specific target entropy settings. The optimal performance is achieved when the targets are set around $H_s = 0.1$ and $H_l = 0.2$.

| $H_s \setminus H_l$ | **0.1** | **0.2** | **0.5** |
|---|---|---|---|
| **0.1** | 69.7 | 70.1 | 68.5 |
| **0.2** | - | 67.7 | 66.3 |
| **0.5** | - | - | 62.1 |

Table 6: Sensitivity analysis in target entropy on the BFCL benchmark. Values represent Overall Accuracy.

**Sensitivity to Entropy Penalty** Table 7 demonstrates the sensitivity analysis regarding the entropy penalty coefficients. The model exhibits considerable stability with respect to $\beta_l$, while lower values of $\beta_s$ generally yield better performance.

| $\beta_s \setminus \beta_l$ | **0** | **e-1** | **e-2** | **e-3** |
|---|---|---|---|---|
| **0** | 70.7 | 68.4 | 71.4 | 70.6 |
| **e-1** | 68.8 | 70.1 | 69.5 | 68.7 |
| **e-2** | - | - | 69.3 | - |
| **e-3** | - | - | - | 70.44 |

Table 7: Sensitivity analysis in entropy penalty on the BFCL benchmark. Values represent Overall Accuracy.

### D.3 GENERALIZATION ACROSS MODEL ARCHITECTURES AND SIZES

To verify the robustness and generalizability of our proposed method, AutoTool, we conducted extensive evaluations across diverse model architectures and parameter scales. Specifically, we utilized the Llama 3.2-inst and Qwen 2.5-inst series, covering a wide range of sizes from 1.5B to 32B parameters.

Figure 11 illustrates the Overall performance on BFCL for both the base models and our method. The visual comparison demonstrates that AutoTool consistently improves performance across all tested configurations.

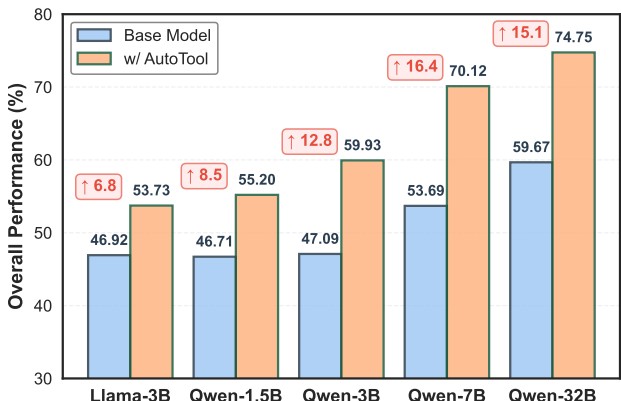

Figure 11: Performance comparison across different model architectures and sizes on BFCL benchmark.

**Key Observations:**

- **Substantial Gains in Qwen Series:** The Qwen models exhibited the most significant improvements. As shown in the figure, models in the 3B to 32B range achieved accuracy increases of over 12 points. For instance, the Qwen2.5-7B model saw a remarkable lift from 53.69 to 70.12.

- **Effectiveness on Compact Models:** Even the smallest tested model (1.5B) demonstrated a substantial performance lift (approx. 8.5 points), confirming the method's efficacy for resource-constrained scenarios.

- **Consistency:** The Llama 3.2 architecture also benefited from our approach, with the 3B model improving from 46.92 to 53.73.

These findings confirm the generalizability and effectiveness of AutoTool across diverse model landscapes, validating that the adaptive mechanism is not limited to a specific architecture.

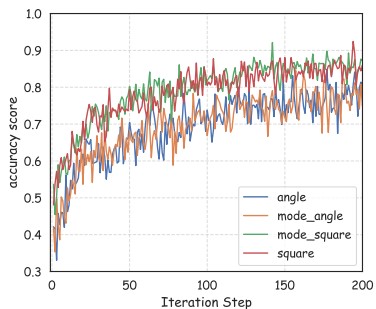 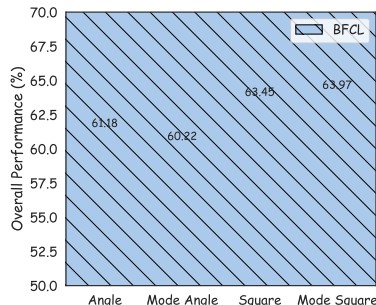

(a) Accuracy Reward by Iteration Step     (b) Performance Comparison of Four Prompt Design

Figure 12: Visualization of training processes and evaluation results for four prompt designs.

## E  PROMPT DESIGN FOR AUTO THINK

To explore a suitable prompt design for Auto Think, we conducted a preliminary analysis of the four kind of prompts listed below:

- Controlled reasoning mode with square tags: **[mode]no_think[/mode] [no_think] \n [/no_think] [tool_call] tool calls here [/tool_call]**

- Uncontrolled reasoning mode with square tags: **[no_think]\n[/no_think] [tool_call] tool calls here [/tool_call]**

- Controlled reasoning mode with angle tags: **<mode>no_think</mode> <no_think> \n </no_think> <tool_call> tool calls here </tool_call>**

- Uncontrolled reasoning mode with angle tags: **<no_think>\n</no_think> <tool_call> tool calls here </tool_call>**

We trained the model starting from Qwen2.5-7B-Instruct using the original GRPO algorithm with PubTool LRL data. Their training processes and evaluation results are presented in Figure 12. From the results, two key observations emerge: (1) Square tags ([]) exhibit better adaptability than angle tags (<>). This may be because the model used angle tags for segmentation in the pre-training phase, reusing these tags for a different purpose (reasoning mode control) is likely to cause signal interference. (2) Additionally, the explicit "reasoning mode" prefix does not obviously affect performance. The evaluation results show that the controlled reasoning mode with square tags achieves the best performance; thus, we adopt this prompt design for auto scaling.

## F  LIMITATION AND FUTURE WORK

Despite promising results in tool-use scenarios, our method has latent concerns to clear. First, we only tested it on a specific model size, future work should verify its scalability across different model scales (e.g., 3B, 13B, 32B parameters) and architecture series, e.g., LLaMA-series. Second, our method's generalizability beyond tool-use tasks is unproven. It is valuable to evaluate its performance on other complex reasoning tasks (e.g., mathematics, logical deduction) to confirm if it can similarly enhance reasoning steps or reduce computational costs. Third, our method currently relies on a specific RL algorithm. Future research should test its compatibility with other RL algorithms (e.g., PPO (Schulman et al., 2017; Engstrom et al., 2020) and DAPO (Yu et al., 2025)) to verify if the decoupled entropy constraint strategy is effective across different algorithmic paradigms. We will address these limitations in future work.

---

**The Full System Prompt for Automatic Think in RL**

You are an advanced function composition agent. Your goal is to solve user queries efficiently. In the multi-turn dialogue loop: the interaction is a cycle: You [tool_call], you receive a [tool_response], and you MUST use that new information to plan your next step.

# Tools
You are provided with function signatures within [tools] and [/tools] tags:
[tools] {functions} [/tools]

# Action Phase
1. Choose an Action Mode: For every turn, you MUST start your response by choosing an action mode (think vs no_think) based on the task's complexity.
    - think: Use for complex reasoning. Enclose your detailed thought process within [think] and [/think] tags.
    - no_think: Use for simple, straightforward tasks. You MUST use an empty, self-closing block: [no_think]\n[/no_think].
Your response MUST begin by enclosing the selected mode name within [mode] and [/mode] tags.

2. Decide on an Action Path: After that, you must choose ONE of the following two paths:

## Path A: Call Functions
WHEN: The user's intent is tool-related and you have all required functions and parameters.
The tool_calls field is a JSON object with function names and arguments within [tool_call] and [/tool_call] XML tags. i.e., [tool_call] [{"name": <function-name>, "arguments": <args-json-object>}, {"name": <function-name2>, "arguments": <args-json-object2>}, ...] [/tool_call]

EXAMPLE:
[mode]no_think[/mode] [no_think]\n[/no_think] [tool_call] tool calls here [/tool_call]
EXAMPLE:
[mode]think[/mode] [think] reasoning process here [/think] [tool_call] tool calls here [/tool_call]

## Path B: Respond Directly to the User
WHEN: You need to provide a natural language text response. This happens in three main scenarios:
(1) After receiving tool execution feedback enclosed within [tool_response] and [/tool_response] tags, continue to respond to user queries based on this feedback.
(2) The user's query is general conversation and not related to any tool.
(3) Ask for more information if the given conversational context lacks the required functions or parameters.

EXAMPLE:
[mode]no_think[/mode] [no_think]\n[/no_think] natural language sentences you talk with user
EXAMPLE:
[mode]think[/mode] [think] reasoning process here [/think] natural language sentences you talk with user

---

Figure 13: System Prompt Design for Automatic Scaling Tool-Use in Multi-Turn Dialogue.

