# OpenReview forum: "AutoTool: Automatic Scaling of Tool-Use Capabilities in RL via Decoupled Entropy Constraints"
_ICLR.cc/2026/Conference — ICLR 2026 Poster_

### Official Review · Reviewer_LhKY · 2025-10-20

**Soundness:** 3
**Presentation:** 3
**Contribution:** 3
**Rating:** 8
**Confidence:** 4

**Summary:**

The paper introduced an adaptive entropy constraint during RL training phase to address the reasoning collapse issue in the hard tool-use task and overthink in the easy cases.

* It provides an new entropy constraint to adaptively prompt long or short reasoning trajectories based on the task difficulty.
* It proposes to use a dataset with emphasis on the difficult task to improve the model performance.
* It provides a thorough analyze on the approach's performance against different approaches proposed to improve tool-use performance and an ablation study on the contribution from different parts of the proposed approach.

**Strengths:**

* Overall. the paper is well-written, it provides an clear illustration on the proposed approach. The experiments setup and results are presented clearly.
* The proposed approach is easy to follow and implement. It provides detailed information on replicating the results.
* The proposed approach improves the model performance on the multi-turn scenario significantly, proving the promoting long reasoning strategy is beneficial to solving complex multi-turn task.
* The ablation study demonstrates the importance of all 3 parts of the proposed approach.

**Weaknesses:**

* While the author puts testing the approach in different architecture series in the future work, but I personally would prefer includes it in the experiments to prove the proposed methods works not just on the Qwen2.5-7B-Instruct.
* Missing how $H_l = 0.2$ and $H_s = 0.1$ are defined, such as a experiment on the performance with different choices of $H_l$ and $H_s$.
* The ablation experiment w/o data refinement does not describe what's the data used in the experiment.
* Fig 4, the loss for short trajectories uses the long trajectory's target entropy.
* It's unclear how are the short and long path/trajectory defined?

**Questions:**

* Line 075, what's the in-depth experiments.
* Line 133, the citation is on the open-source code which does not provide useful information on the finding.
* Table 2, why is the # of data in ToolACE and Hermes Function-Calling remain the same after downsampling.
* Sec 4.3.2, if the ACU includes the accuracy and # of tokens, shouldn't the ACU score be the overall performance metric?
* Line 752, it states `Results indicate data refinement increases accuracy reward score by +15%`, but the figure 9 shows the other way around. The blue line with data refinement does not have much accuracy score increment.
* Line 911, PPO citation incorrect?
* While the # of turns might correlate to the task difficulty, but it would be nice to have an analyze on the performance (accuracy, # of tokens) in different task difficulty.

---

> ### Author Response · Authors · 2025-11-25
>
> Thank you for your detailed review and approval of this work, and for providing this amount of review comments.
>
> We will address your confusion one by one.

---

> ### Author Response · Authors · 2025-11-25
> **Response to all Questions**
>
> **Response to Question 1**: It refers to analysis study in Section 2.
>
> **Response to Question 2**: We cite *open-r1* code repository by huggingface, is the first open-source framework to successfully reproduce DeepSeek-R1’s training pipeline. Other concurrent works (e.g., [1, 2]) report similar findings.
>
> [1] 7B Model and 8K Examples: Emerging Reasoning with Reinforcement Learning is Both Effective and Efficient
>
> [2] Simplerl-zoo: Investigating and taming zero reinforcement learning for open base models in the wild
>
> **Response to Question 3**: As stated in Section 3.1 (P1), downsampling is applied to balance the dataset, however, the two smaller datasets are excluded from downsampling due to their limited sample sizes.
>
> **Response to Question 4**: Yes, ACU can be the overall performance metric.
>
> **Response to Question 5**: We apologize for the legend error in Figure 9: the blue line should corresponds to w/o data refinement, and the orange line to w/ data refinement. Thank you for catching this, we have corrected it in the revision.
>
> **Response to Question 6**: Yes, thank you for pointing this out. We have corrected the citation in the revision:
> [1] Schulman et al., Proximal Policy Optimization Algorithms, 2017.
> [2] Engstrom et al., Implementation Matters in Deep Policy Gradients: A Case Study on PPO and TRPO
>
> **Response to Question 7**: As suggested, we analyzed how accuracy and token usage vary with the number of turns.
>
> Experimental Results:
> | # Turn  | Acc  | # Tokens |
> | ------------ | ------------ | ------------ |
> | 1  | 84.99  | 80.85  |
> | 2  | 49.35  | 232 |
> | 4  | 43.35  | 532 |
> | 6  | 36.32  | 764 |
> | 8 | 33.32  | 1094 |
> | 10+ | 19.65  | 1435|
>
> Observation: As the number of turns increases, we observe a clear downward trend in accuracy combined with an upward trend in token costs. This indicates that for more complex tasks (which require more turns), the model’s performance degrades even as it generates significantly more tokens in its attempt to solve the problem.

---

> ### Author Response · Authors · 2025-11-25
> **Response to Weakness 3-5**
>
> **Response to Weakness 3:** The ablation experiment w/o data refinement is conducted on the curated PubTool dataset; we have cleared it in Section B
>
> **Response to Weakness 4**: Indeed, thank you for catching this notation error; we have corrected it in the revised manuscript.
>
> **Response to Weakness 5**: We introduce long-path and short-path responses to denote outputs with and without explicit reasoning, respectively.  We have cleared this define in Introduction in revised manuscript.

---

> ### Author Response · Authors · 2025-11-26
> **Response to Weakness 1**
>
> Sure.
> We evaluated our approach, AutoTool, across various model architectures and sizes, specifically utilizing the Llama 3.2 and Qwen 2.5 series (ranging from 1.5B to 32B parameters).
>
> As shown in the table below, our method consistently improved performance across all tested models and sizes. Notably, AutoTool yielded substantial gains:
> - The Qwen models showed the most significant improvement, with the 3B to 32B sizes achieving an increase of over $12\%$.
> - The 1.5B size model saw a lift of $9.5\%$.
> - The Llama model demonstrated a performance gain of $8.2\%$.
>
> These findings confirm the generalizability and effectiveness of our method across diverse model landscapes.
>
> | Model  | Overall Acc  |  Model  | Overall Acc |
> | ------------ | ------------ | ------------ | ------------ |
> | llama3.2-3B-Instruct  | 46.92 | Qwen2.5-1.5B-Instruct   | 46.71  |
> | ~ w/ AutoTool  | 53.73  | ~ w/ AutoTool  | 55.20  |
> | Qwen2.5-7B-Instruct  | 53.69 | Qwen2.5-3B-Instruct   | 47.09  |
> | ~ w/ AutoTool  | 70.12  | ~ w/ AutoTool  | 59.93  |
> | Qwen2.5-32B-Instruct   | 59.67 | -   | -  |
> | ~ w/ AutoTool  | 74.75 | -  | - |

---

> ### Author Response · Authors · 2025-11-26
> **Response to Weakness 2**
>
> In our manuscript, target entropy values ($H_s$ and $H_l$ ) were initially selected based on empirical observation (Figure 2-d). Acknowledging the heuristic nature of this choice, we conducted a comprehensive ablation study over entropy targets.
>
> We evaluated the performance on the BFCL benchmark, reporting the Overall Accuracy as shown in the table below. The results indicate that model performance is highly sensitive to the specific target entropy values. The optimal performance is achieved when the targets are set around $H_s = 0.1$ and $H_l = 0.2$
>
> | $H_s$\ $H_l$  | 0.1  | 0.2  | 0.5  |
> | ------------ | ------------ | ------------ | ------------ |
> | 0.1  | 69.7  | 70.1  | 68.5  |
> | 0.2  | -  | 67.7  | 66.3  |
> | 0.5  | -  | - | 62.1  |

---

> ### Author Response · Authors · 2025-11-28
> **Follow-Up on Responses**
>
> Dear Reviewer,
>
> Thank you again for acknowledging the contributions of this work.
>
> With the discussion period closing soon, we would like to respectfully follow up and inquire if you have the opportunity to review our responses to your comments.
>
> We sincerely look forward to your feedback.

---

### Official Review · Reviewer_nwbB · 2025-10-31

**Soundness:** 3
**Presentation:** 3
**Contribution:** 3
**Rating:** 6
**Confidence:** 4

**Summary:**

The paper proposes AutoTool, an RLVR-style training recipe for tool use that decouples entropy regularization between short (no-think) and long (think) trajectories and adapts the long-path entropy coefficient toward a target entropy. The pipeline warms up with SFT to expose both modes, then runs GRPO with: (i) a mode tag ([think]/[no_think]) and a format reward, (ii) an answer reward, and (iii) an entropy loss that activates only when entropy drops below a target (adaptive β for think, fixed β for no-think). On BFCL (non-live, live, multi-turn), API-Bank, and ACEBench, a ~7B model improves accuracy vs. SFT/GRPO baselines while cutting inference tokens, and the model auto-scales its "think rate".

**Strengths:**

1. Clear motivation & mechanism. The paper links tool-use failures under RL to entropy collapse and proposes a targeted fix—decoupled, adaptive entropy—implemented with minimal changes to GRPO.
2. Solid empirical suite. Evaluations span BFCL (non-live/live/multi-turn), API-Bank, and ACEBench, with consistent gains over SFT/distillation/RLVR-like baselines; ablations remove each component (data refine, decouple, adapt-coef) and show clear drops.
3. Efficiency analysis. The paper reports token-cost reductions, defines ACU (Accuracy per Params×Tokens), and visualizes training dynamics (entropy, think-rate, lengths).

**Weaknesses:**

1. Data construction bias and potential leakage. The curated training set mixes downsampled public data with RL-refined, low-variance samples, which likely skews toward cleaner/easier cases; possible overlaps with evaluation sets are not audited.
2. Hyperparameter sensitivity. In multi-turn settings, performance appears sensitive to target entropies and the choice of β; broader sweeps or learned schedules would strengthen the claims.

**Questions:**

1. Sensitivity & ablations. Will you provide sweeps over target entropies for think/no-think and over β (fixed vs. adaptive), and include a single unified entropy-penalty baseline to isolate the benefit of decoupling?
2. Format reliance. Can you evaluate without the [think]/[no_think] tags (no format reward), or on prompts that prohibit tags, to test format-agnostic generalization?
3. Data & overlap checks. Would you release the curated training data (or a representative subset) and report an overlap analysis with evaluation sets, and add results where any dataset family used for training is excluded from the evaluation domains?

---

> ### Author Response · Authors · 2025-11-25
>
> Thank you for your valuable feedback.
>
> We will address your confusion one by one.

---

> ### Author Response · Authors · 2025-11-25
> **Response to Weakness 1 & Question 3**
>
> we will fully release the curated training data upon acceptance.
>
> Moreover, we have conducted a thorough overlap analysis between the training and evaluation sets, reporting results across two distinct metrics:
> - Question Overlap: The ratio of identical user questions (based on exact string match).
> - Tool Overlap: The ratio of candidate tools shared between the datasets (based on unique tool names).
>
> | Set Name | # Question | overlap |
> | :--- | :--- | :--- |
> | PubTool | 27429 | 0% |
> | BFCL | 4072 | 0% |
>
> | Set Name | # Tool | overlap |
> | :--- | :--- | :--- |
> | PubTool | 8855 | 2.51% |
> | BFCL | 2031 | 2.51% |
> The results show zero question overlap and only a minimal tool overlap (2.51%) across the datasets.
>
> Third, we also add results where any dataset family used for training is excluded from the evaluation domains.
> Since the question overlap is $0\%$, we removed all training examples that contained any of overlapping tools to create a "tool-overlap-free" training set.
> The results are presented below:
>
> |   |  Non-Live | Live | Multi-Turn | Overall Acc|
> | ------------ | ------------ | ------------ | ------------ | ------------ |
> | Qwen2.5-7B-Instruct |86.46 | 67.44 | 7.62 | 53.69 |
> | AutoTool-7B | 89.76  | 80.22  |  38.18 | 70.12  |
> | + w/o tool overlap  | 89.85  | 80.02  | 36.41  | 68.76 |
> | Change | +0.09|	-0.20|	-1.77|	-1.36|
>
> The removal of training data corresponding to the overlapping tools resulted in a negligible performance drop ($\sim1.36\%$ overall). This robust result demonstrates that our approach does not rely on memorizing specific tool instances and highlights its tool-independent generalization.

---

> ### Author Response · Authors · 2025-11-26
> **Response to Question 2**
>
> Sure.
> We evaluated the extent of format reliance by removing the explicit format reward.
> We also compared the performance using other types of tags, as detailed in Section E.
>
> | Format Types | Overall Acc   |  Format Types | Overall Acc |
> | ------------ | ------------ | ------------ | ------------ |
> |  Angle | 66.51  |   Mode Angle |66.35     |
> | Square | 69.41 | Mode Square | 70.12
> | w/o reward | 64.75 | - | -|
>
> As shown in the table above, the absence of the format reward led to a performance drop of 1.76% to 5.37% (e.g., 66.51% to 64.75%). These findings demonstrate that an explicit format constraint is essential for improving reasoning performance.

---

> ### Author Response · Authors · 2025-11-26
> **Response to Weakness 2 & Question 1**
>
> Of course.
>
> To end this, we conducted a comprehensive sensitivity study on the key hyperparameters, including the target entropies ($H_s$ and $H_l$) and the entropy penalties ($\beta_s$ and initial $\beta_l$). We evaluated the performance on the BFCL benchmark, reporting the Overall Accuracy, as shown in Table 1 & 2.
> Also, for comparison, we also assessed baselines utilizing a single, unified entropy penalty ($\beta$) in the RLVR setting, as detailed in Table 3.
>
> Table 1: Sensitivity analysis in target entropy
> | $H_s$\ $H_l$  | 0.1  | 0.2  | 0.5  |
> | ------------ | ------------ | ------------ | ------------ |
> | 0.1  | 69.7  | 70.1  | 68.5  |
> | 0.2  | -  | 67.7  | 66.3  |
> | 0.5  | -  | - | 62.1  |
>
> Table 2: Sensitivity analysis in entropy penalty
> | $\beta_s$\ $\beta_l$ | 0  | e-1  | e-2  |  e-3 |
> | ------------ | ------------ | ------------ | ------------ | ------------ |
> | 0  | 70.7  | 68.4  | 71.4  | 70.6  |
> | e-1 | 68.8  | 70.1  | 69.5  | 68.7  |
> | e-2  | - | - | 69.3  | -  |
> | e-3  | -  | -  | -  | 70.44  |
>
>
> Table 3: Comparison with unified entropy penalty baselines
> |  Methods | Non-Live | Live | Multi-Turn | Overall Acc|
> | ------------ | ------------ | ------------ | ------------ | ------------ |
> |  PubTool-GRPO |   88.87  | 78.93  | 10.77   | 60.13    |
> | w/ entropy penalty   |   |   |   |   |
> |  + $\beta=e-1$ | 66.36  | 62.32  | 6.12  | 44.93  |
> |  + $\beta=e-2$ | 71.32  | 66.37  |  7.95 | 48.54  |
> |  + $\beta=e-3$ | 89.35  | 80.81  | 13.61  | 61.25  |
> | AutoTool(Our)  | 89.76  | 80.22  | 38.18  | 70.12  |
>
> From Table 1, model performance is sensitive to the specific target entropy values.
> The optimal performance is achieved when the targets are set around $H_s = 0.1$ and $H_l = 0.2$.
>
> Table 2 demonstrates that the model exhibits considerable stability with respect to $\beta_l$, while lower values of $\beta_s$ yield better performance.
>
> From Table 3，The entropy penalty demonstrated a slight improvement (from 60.13 to 61.25 with $\beta=e-3$) in performance only under a specific, finely-tuned penalty weight. This highlights that simpler constraints are generally ineffective or highly sensitive to hyperparameter choices.

---

> ### Author Response · Authors · 2025-11-28
> **Follow-Up on Responses**
>
> Dear Reviewer,
>
> Thank you again for acknowledging the contributions of this work.
>
> With the discussion period closing soon, we would like to respectfully follow up and inquire if you have the opportunity to review our responses to your comments.
>
> We sincerely look forward to your feedback.

---

### Official Review · Reviewer_URMW · 2025-10-31

**Soundness:** 1
**Presentation:** 1
**Contribution:** 1
**Rating:** 2
**Confidence:** 4

**Summary:**

This paper aims to improve Tool-use LLMs' performance through the control of thinking length. It proposes applying distinct entropy constraints to long and short trajectories during RL.

**Strengths:**

* This paper proposes a new loss function designed to apply an entropy penalty that is conditional on the trajectory length within the RL process.

**Weaknesses:**

* The paper suffers from a critically confusing narrative and fundamental conceptual ambiguity. The core concept of TTS introduced in line 40 is unsupported in the literature and largely diverges from the definitions provided in the very papers cited. This narrative choice, which mixes TTS terminology with RL problem, is highly confusing and obscures the paper's actual contribution.
* For the RL component, the idea of activating or deactivating thinking via special tokens is a well-established and standard paradigm. The paper's contribution is reduced to proposing a new training loss within this existing framework, making the technical novelty incremental. The authors fail to properly contextualize their work within the relevant RL literature, which is a major omission.
* The experiments also lack rigor.
  * The paper largely ignores previous work on critical related issues, such as thinking length and entropy constraints in RLVR.
  * The evaluation is demonstrably unfair due to a vast disparity in the inference budget. Figure 1-c explicitly shows the proposed method uses an inference budget greater than 1k, while the baselines are restricted to less than 100. This makes the reported performance gains unconvincing, especially since many *actual* TTS methods can achieve similar gains without requiring any additional RL training.

**Questions:**

See above.

---

> ### Author Response · Authors · 2025-11-25
>
> Thank you for your valuable feedback.
>
> We will address your confusion one by one.

---

> ### Author Response · Authors · 2025-11-25
> **Response to Weakness 1**
>
> There is a confusion.
> The core concept of TTS definition is supported in the two cited literature and not largely diverges from the definitions in cited literature.
>
> literature 1: *s1: Simple test-time scaling*.
> It defines test-time scaling  as a promising new approach to language modeling that uses extra test-time compute to improve performance(1-2 line). There are many techniques to increase the compute at test time to get better results,  Monte Carlo Tree Search, multi-agent approache, long CoT (scale up explicit reasoning length in response). In this literature, it instantiate TTS by explicitly lengthening the model’s reasoning length in its responses.
>
> literature 2: *DeepSeek-R1: Incentivizing Reasoning Capability in LLMs via Reinforcement Learning*.
> It demonstrates, for the first time, that extending reasoning length at inference (via RL) serves as an effective instantiation of TTS, even without coining the term explicitly (Sec. 3, Para. 2).
>
> Overall, scaling up response length via reinforcement learning (RL) is a dominant paradigm for achieving test-time scaling; our work builds directly upon this established approach.
>
> To clear your confusion and avoid ambiguity , we have revised the text to define TTS as:
>
> `Test-time scaling is a approach to language modeling that uses extra test-time compute to improve performance. Scaling up a model’s explicit reasoning length via RL is a effective way to achieve TTS.`

---

> ### Author Response · Authors · 2025-11-25
> **Response to Weakness 2**
>
> While manually toggling reasoning via special tokens is well-established (e.g., Qwen3, GLM-4.5), automatically adjust reasoning scale, especially in agentic tool use, remains largely unexplored. **To our knowledge, our work is the first to investigate adaptive reasoning strategies in agentic tool use**.
>
> **We would like to respectfully point out that the technical novelty of this work is not incremental, but substantial and is strongly supported by the other three reviewers.**
> For instance, Reviewer dVK5 explicitly states that our decoupled entropy constraint mechanism is novel, and Reviewer nwbB notes that our approach (minimal changes to GRPO) demonstrates a solid empirical performance.
> In our view, proposing a simple yet effective improvement for an existing training algorithm can be just as crucial for driving technical progress as introducing a completely new one.
>
> Due to page limitation, in our source manuscript, we contextualize these prior works in Section A (Appendix).

---

> ### Author Response · Authors · 2025-11-26
> **Response to Weakness 3**
>
> **Response to 3.A:**
>
> Thank you for your concerns about the previous work experiment.
> To address your concerns, we have expanded the study as follows.
>
> Given that we are the first to investigate adaptive reasoning in agentic tool use, there is a significant lack of prior works on this domain, we compare our method with recent adaptive reasoning strategies adapted from other domains (e.g., math). We select two representative methods for comparison:
>
> - [1] Thinkless: LLM Learns When to Think (2025-05): Achieves adaptive reasoning via a control token loss.
> - [2] Adactrl: Towards adaptive and controllable reasoning via difficulty-aware budgeting(2025-05): The model self-assesses problem difficulty during the RLVR rollout to facilitate adaptive reasoning.
>
> We replicated both methods using the same dataset (PubTool) and the same base model (Qwen2.5-7B-Instruct). The experimental results are summarized in Table 1.
> The experimental results are summarized in Table 1.
>
> Table 1: Comparison with other Adaptive Reasoning Methods
> |  Methods | Non-Live | Live | Multi-Turn | Overall Acc|
> | ------------ | ------------ | ------------ | ------------ | ------------ |
> |  Base |   86.36  | 67.44  | 7.62   | 53.69    |
> |  PubTool-GRPO |   88.87  | 78.93  | 10.77   | 60.13    |
> |  Adactrl |   86.36  | 73.12  | 15.63  |  58.37  |
> |  Thinkless  | 86.92  | 77.62  | 24.64  | 63.06  |
> | AutoTool(Our)  | 89.76  | 80.22  | 38.18  | 70.12  |
>
> Furthermore, we investigated the effect of incorporating simple normal entropy and length constraints as baselines, where we penalize excessively low entropy and low thinking lengths, respectively. The experimental results are summarized in Table 2.
>
> Table 2: The thinking length and entropy constraints study in RLVR
> |  Methods | Non-Live | Live | Multi-Turn | Overall Acc|
> | ------------ | ------------ | ------------ | ------------ | ------------ |
> |  PubTool-GRPO |   88.87  | 78.93  | 10.77   | 60.13    |
> | w/ entropy constraint   |   |   |   |   |
> |  + $\beta=e-1$ | 66.36  | 62.32  | 6.12  | 44.93  |
> |  + $\beta=e-2$ | 71.32  | 66.37  |  7.95 | 48.54  |
> |  + $\beta=e-3$ | 89.35  | 80.81  | 13.61  | 61.25  |
> | w/ length constraint   |   |   |   |   |
> |  + $l=100$ | 88.54  | 77.65  | 8.82  | 58.37  |
> |  + $l=50$ | 87.22  | 77.96  | 10.12  | 58.43  |
> | AutoTool(Our)  | 89.76  | 80.22  | 38.18  | 70.12  |
>
> The results in Table 1 demonstrate the superiority of our AutoTool method. Specifically, AutoTool outperforms the best prior method (Thinkless) by a significant margin of 7.06 percentage points. Adactrl performs worse than native GRPO, suggesting this strategy is unsuitable for agentic tool use tasks.
> The results in Table 2, The simple length constraint did not exhibit positive effects when applied to the PubTool-GRPO baseline.
> The entropy constraint demonstrated a slight improvement (from 60.13 to 61.25 Overall Accuracy) in performance only under a specific, finely-tuned penalty weight.
> This highlights that simpler constraints are generally ineffective or highly sensitive to hyperparameter choices.
>
> **Response to 3.B:**
>
> There appears to be a misunderstanding: in Figure 1-c, the token cost refers to the actual output length, not the inference budget. In our experiments, all models share the same inference-length budget. However, non-reasoning models (e.g., base or sft model) often terminate early (e.g., by emitting stop tokens), yielding shorter reasoning traces; in contrast, scaled-up models fully utilize the budget and generate substantially longer, more complete reasoning chains.

---

> ### Author Response · Authors · 2025-11-27
> **Follow-Up on Responses**
>
> Dear Reviewer,
>
> Thank you for your efforts on our submission.
> As the rebuttal period come to end, we would like to respectfully follow up and inquire if you have the opportunity to review our responses to your comments.
> We have conducted extensive additional experiments to address your concerns.
>
> We would greatly appreciate your consideration in adjusting the scores accordingly.

---

> ### Author Response · Authors · 2025-11-28
> **Follow-Up on Responses**
>
> Dear Reviewer URMW,
>
> Thank you for your effort. We are writing to respectfully inquire if you have had the opportunity to review our response.
>
> There is no fatal weaknesses in the manuscript, and we confirm that all your concerns have been thoroughly addressed:
> 1. The terminological imprecision has been revised and clarified.
> 2. Novelty concerns have been explained and are supported by other reviewers' comments.
> 3. We have added more previous works as baselines to strengthen our empirical conclusions.
>
> We emphasize the core contribution: This work makes a **crucial first step** in realizing **auto-scaling in tool use**, which simultaneously improves accuracy and significantly reduces inference overhead. **This is a clear and genuine advance for the research field.**
> Therefore, We believe that this work should not be rejected.
>
> We sincerely appreciate your time and consideration in re-evaluating your assessment and adjusting the scores accordingly.

---

### Official Review · Reviewer_dVK5 · 2025-11-06

**Soundness:** 3
**Presentation:** 4
**Contribution:** 3
**Rating:** 6
**Confidence:** 4

**Summary:**

This paper proposes a novel reinforcement learning training paradigm to address two key challenges in tool-based AI agents: (1) direct RL training struggles to scale up thinking length sufficiently to solve complex problems, and (2) scaled-up models tend to overthink simpler problems, resulting in substantial token inefficiency.

Main contributions include:
1. Proposed a decoupled adaptive entropy constraint strategy, which separates entropy constraints for short and long reasoning paths, enabling the model to automatically adjust its reasoning scale based on problem difficulty.
2. Designed an automatic test-time scaling reward module, which uses an asymmetric reward mechanism to encourage short reasoning for simple problems and long reasoning for complex ones.
3. Constructed the PubTool dataset and optimized RL training effectiveness through a data quality refinement strategy.
4. Validated the method's effectiveness on three benchmarks, achieving a significant 9.8% accuracy improvement while reducing computational overhead by approximately 81%.

**Strengths:**

- The decoupled entropy constraint mechanism is novel. By applying different entropy constraint strengths to short and long reasoning paths, it effectively solves the reasoning collapse problem prevalent in traditional RL training.
- The paper is well-structured, with logical coherence from problem analysis and method design to experimental validation. The figures are well-designed, particularly the training dynamics visualization, which effectively demonstrates the auto-scaling effects.
- The experimental design is comprehensive, featuring systematic evaluation on three benchmarks: BFCL, API-Bank, and ACEBench.

**Weaknesses:**

- Entropy constraint hyperparameter selection: Although an adaptive mechanism is proposed, the initial choices for H_l and B_s lack sufficient theoretical justification or ablation analysis.
- While sample filtering based on reward variance is mentioned, the specific threshold settings and selection criteria are not described in detail.

**Questions:**

- The legend for Figure 2 could be added to each of the three subplots or provided as a common legend separately.
- What is the specific theoretical rationale behind the asymmetric reward in Formula (5) (only +0.5 for correct answers in 'think' mode versus +1.0 in 'no-think' mode)? Could this design potentially bias the model towards preferring the 'no-think' mode?

---

> ### Author Response · Authors · 2025-11-25
>
> Thank you for your valuable reviews.
>
> We will address your confusion one by one.

---

> ### Author Response · Authors · 2025-11-25
> **Response to Weakness2 & all Questions**
>
> **Response to Weakness 2**: To align the dataset size with that of SFT (~8K samples), we rank examples by reward variance and filter out the bottom 55%, a simple yet effective heuristic to retain higher-signal instances while controlling scale.
>
> **Response to Question 1**: Thank you for the suggestion. in Reversion, we have added the legend to each subplot in Figure 2.
>
> **Response to Question 2**: The asymmetric reward (+1.0 for correct no-think answers vs. +0.5 for correct think answers) incentivizes the model to answer directly when confident, achieving high accuracy with minimal token cost, while reserving deliberative think mode for challenging questions where exploration is beneficial.
>
> Thus, the design adaptively biases the model toward no-think when capable, and think when uncertain, balancing efficiency and robustness.

---

> ### Author Response · Authors · 2025-11-26
> **Response to Weakness1**
>
> To address this, we conducted a hyperparameter sensitivity analysis to explore their effects, including the target entropies ($H_s$ and $H_l$) and the entropy penalties ($\beta_s$ and initial $\beta_l$).
>
> We evaluated the performance on the BFCL benchmark, reporting the Overall Accuracy, as shown in Table 1 & 2 (due to rebuttal time constraints, part of the results are vacant.).
>
> Table 1: Sensitivity analysis in target entropy
> | $H_s$\ $H_l$  | 0.1  | 0.2  | 0.5  |
> | ------------ | ------------ | ------------ | ------------ |
> | 0.1  | 69.7  | 70.1  | 68.5  |
> | 0.2  | -  | 67.7  | 66.3  |
> | 0.5  | -  | - | 62.1  |
>
> Table 2: Sensitivity analysis in entropy penalty
> | $\beta_s$\ $\beta_l$ | 0  | e-1  | e-2  |  e-3 |
> | ------------ | ------------ | ------------ | ------------ | ------------ |
> | 0  | 70.7  | 68.4  | 71.4  | 70.6  |
> | e-1 | 68.8  | 70.1  | 69.5  | 68.7  |
> | e-2  | - | - | 69.3  | -  |
> | e-3  | -  | -  | -  | 70.44  |
>
> From Table 1, model performance is sensitive to the specific target entropy values.
> The optimal performance is achieved when the targets are set around $H_s = 0.1$ and $H_l = 0.2$.
> Table 2 demonstrates that the model exhibits considerable stability with respect to $\beta_l$, while lower values of $\beta_s$ yield better performance.

---

> ### Author Response · Authors · 2025-11-28
> **Follow-Up on Responses**
>
> Dear Reviewer,
>
> Thank you again for acknowledging the contributions of this work.
>
> With the discussion period closing soon, we would like to respectfully follow up and inquire if you have the opportunity to review our responses to your comments.
>
> We sincerely look forward to your feedback.

---

### Author Response · Authors · 2025-11-27
**General Response to Reviewers & ACs**

Dear Reviewers & ACs,

We sincerely thank all reviewers for their valuable time and effort. We particularly appreciate Reviewer LhKY for their thorough and constructive feedback.

Our work is the first to investigate adaptive reasoning strategies in agentic tool use. We propose a novel decoupled adaptive entropy constraint strategy that automatically adjusts its reasoning scale, which simultaneously improves accuracy and significantly reduces inference overhead.
This study represents a crucial first step toward realizing auto-scaling in tool use, marking a clear and genuine advance for the research field.

We have fully addressed all weaknesses raised by the reviewers, including:
1. Hyperparameter Sensitivity：We have conducted a detailed empirical study to fully justify the chosen hyperparameter settings.
2. Backbone Models Limitation: We extended the implementation across different model series and sizes, demonstrating the adaptability and generalizability of our method.
3. Previous Works Baseline Concerns: We have added more than four kinds of new baselines to further strengthen and support our experimental conclusions.

We thank the reviewers once again for their valuable suggestions, which have greatly helped us enhance the quality and clarity of this paper.

---

### Meta-Review · Area_Chair_iSCi · 2026-01-07

**Summary:**

This paper proposes "AutoTool", an RL-based training framework for agentic tool use that aims to automatically scale reasoning length by 'decoupling' entropy constraints between short (“no-think”) and long (“think”) trajectories. Reviewers broadly agree that the problem (i.e. balancing effective long-horizon reasoning with token efficiency in tool use) is important and timely. The paper presents a coherent method and a fairly extensive experimental evaluation, with several reviewers finding the approach reasonable and empirically effective. However, opinions diverge on the depth of technical novelty, clarity of positioning relative to prior RLVR and adaptive reasoning work, and rigor of experimental comparisons, leading to a borderline overall assessment.

**Reviewer Concerns:**

Concerns fully or mostly addressed by the rebuttal:
* Hyperparameter sensitivity and ablations: The authors added detailed sensitivity analyses for target entropies and entropy penalties, as well as comparisons against unified entropy baselines, addressing a major concern raised by multiple reviewers.
* Baseline coverage and fairness: Additional adaptive-reasoning baselines (e.g., Thinkless, Adactrl) were implemented under matched settings, and the authors clarified confusion around inference budget vs. actual generated token length.
* Data leakage and overlap: The authors provided explicit overlap analyses between training and evaluation data (question- and tool-level), and showed negligible performance degradation when overlapping tools were removed.
* Generalizability across models: Experiments were extended to multiple backbone architectures and sizes (Qwen and Llama families, 1.5B–32B), demonstrating consistent gains.

Concerns only partially or not addressed by the rebuttal:
* Incremental nature of the core contribution: Some reviewers remain unconvinced that the technical advance (decoupled and adaptive entropy constraints within an existing RLVR-style framework) constitutes a sufficiently deep novelty rather than a (well-engineered) extension of known ideas.
* Conceptual framing and clarity: The connection to “test-time scaling” and the broader RL literature could still be clearer, particularly for readers less familiar with recent RLVR and adaptive reasoning work.
* Evaluation scope: While results are consistent across several benchmarks, the experiments remain confined to tool-use tasks; it is still unclear how broadly the proposed mechanism generalizes beyond this setting.

**Reviewer Scores:**

* Reviewer dVK5: Likely remains at 6. This reviewer viewed the entropy-decoupling mechanism as novel and found the experiments convincing; the rebuttal addressed their main technical concerns.
* Reviewer nwbB: Likely remains at 6, possibly goes up to 8. The added overlap checks, ablations, and format-robustness experiments directly addressed their questions.
* Reviewer LhKY: Likely remains at 8. This reviewer already considered the contribution solid, and the rebuttal resolved most clarity issues.
* Reviewer URMW: Should likely increase to 4 after rebuttal if engaging in good faith. While still skeptical about novelty and framing, many of the strongest objections (unfair evaluation, missing baselines, lack of rigor) were directly addressed and do not appear fatal.

Overall:  After accounting for rebuttal-driven adjustments, the score would remain somewhat borderline, slightly tilting towards acceptance.

---

### Decision · Program_Chairs · 2026-01-26

Accept (Poster)